# Synthetic Biology on Acetogenic Bacteria for Highly Efficient Conversion of C1 Gases to Biochemicals

**DOI:** 10.3390/ijms21207639

**Published:** 2020-10-15

**Authors:** Sangrak Jin, Jiyun Bae, Yoseb Song, Nicole Pearcy, Jongoh Shin, Seulgi Kang, Nigel P. Minton, Philippe Soucaille, Byung-Kwan Cho

**Affiliations:** 1Department of Biological Sciences, Korea Advanced Institute of Science and Technology, Daejeon 34141, Korea; powerjin1224@kaist.ac.kr (S.J.); jiyunbae@kaist.ac.kr (J.B.); yosebiback@kaist.ac.kr (Y.S.); joshn@kaist.ac.kr (J.S.); seul9@kaist.ac.kr (S.K.); 2KAIST Institute for the BioCentury, Korea Advanced Institute of Science and Technology, Daejeon 34141, Korea; 3BBSRC/EPSRC Synthetic Biology Research Centre (SBRC), School of Life Sciences, University Park, The University of Nottingham, Nottingham NG7 2RD, UK; Nicole.Pearcy@nottingham.ac.uk (N.P.); nigel.minton@nottingham.ac.uk (N.P.M.); soucaill@insa-toulouse.fr (P.S.); 4Université de Toulouse, INSA, UPS, INP, LISBP, 31400 Toulouse, France; 5Institut National de la Recherche Agronomique (INRA), UMR 792, 31077 Toulouse, France; 6Centre National de la Recherche Scientifique (CNRS), UMR 5504, 31400 Toulouse, France; 7Innovative Biomaterials Center, Daejeon 34141, Korea; 8Intelligent Synthetic Biology Center, Daejeon 34141, Korea

**Keywords:** acetogenic bacteria, C1 gas fixation, synthetic biology, CRISPR-Cas

## Abstract

Synthesis gas, which is mainly produced from fossil fuels or biomass gasification, consists of C1 gases such as carbon monoxide, carbon dioxide, and methane as well as hydrogen. Acetogenic bacteria (acetogens) have emerged as an alternative solution to recycle C1 gases by converting them into value-added biochemicals using the Wood-Ljungdahl pathway. Despite the advantage of utilizing acetogens as biocatalysts, it is difficult to develop industrial-scale bioprocesses because of their slow growth rates and low productivities. To solve these problems, conventional approaches to metabolic engineering have been applied; however, there are several limitations owing to the lack of required genetic bioparts for regulating their metabolic pathways. Recently, synthetic biology based on genetic parts, modules, and circuit design has been actively exploited to overcome the limitations in acetogen engineering. This review covers synthetic biology applications to design and build industrial platform acetogens.

## 1. Introduction

Acetogenic bacteria (acetogens) reduce C1 gases such as CO_2_ and CO, into acetyl-CoA using the Wood-Ljungdahl (WL) pathway, and acetyl-CoA is ultimately converted into acetate. Over 100 acetogen species from 20 different genera have been isolated from diverse habitats, making the bacteria phylogenetically and physiologically diverse with a wide range of optimal growth conditions [1,2,3,4,5]. Diverse acetogens have been suggested as promising biocatalysts to utilize C1 gases in synthesis gases or waste gases generated from industries using the WL pathway. However, despite their high industrial potential, their commercialization on the industrial scale is limited because of their slow growth rates and low productivity when using C1 gases as their sole carbon and energy sources. In addition, the lack of genetic manipulation tools also makes it challenging to apply acetogens at the industrial scale [6,7,8,9,10].

To overcome these hurdles, synthetic biology provides many ways to build engineered acetogens characterized by optimal growth rates and maximal productivities. To this end, it is necessary to analyze their metabolic pathways and energy conservation systems, along with the development of highly efficient synthetic bioparts and genetic manipulation tools. Eventually, these efforts will build various genetic circuits capable of producing predictable outputs in response to various input signals through standardized components and modularization [11,12,13,14,15]. In recent years, various microorganisms have been engineered to maximize their productivity by establishing predictable systems through the development of synthetic promoters, untranslated regions (UTRs), evolved metabolic enzymes, genome-editing tools, regulatory circuits, and chassis platforms [11,12,16,17,18,19,20,21,22]. If these synthetic biology approaches are fully applied to acetogens, many of their limitations, such as the ones mentioned above, could be resolved. This review summarizes the metabolic engineering efforts on acetogens so far and introduces the synthetic biology approaches required for efficient C1-to-biochemical conversion.

## 2. Development of Genetic Manipulation Tools in Acetogens

Acetogens have high potential as biocatalysts, but there have been many difficulties in their industrial application due to the limited availability of genetic manipulation tools. This is because most acetogens are Gram-positive bacteria and have a thick cell wall, which makes them recalcitrant to receiving foreign DNA molecules [10,23,24]. Nevertheless, plasmid-based gene expression methods developed for *Clostridium* species have been shown to be functional in other acetogens. In addition, many metabolic engineering efforts have recently been made using homologous recombination (HR) [25,26,27,28,29,30,31,32,33,34,35], ClosTron [28,36,37,38,39] and Clustered Regularly Interspaced Short Palindromic Repeats (CRISPR) along with its CRISPR-associated (Cas) protein (CRISPR-Cas) system [32,40,41,42,43,44,45] to improve the production of value-added biochemicals from C1 gases.

### 2.1. Development of Plasmid-Based Engineering Tools in Acetogens

To express foreign DNA in acetogens, a plasmid was first constructed in *E. coli* and then transferred to acetogens. Therefore, many vectors are shuttle vectors, with two replication origins for *E. coli* and the target acetogen (Table 1). The pIMP1 vector usable in *Clostridium aceticum*, *Clostridium ljungdahlii*, and *Clostridium autoethanogenum* consists of pIM13 of gram-positive origin and ColE1 of gram-negative origin. Since *ermC* encodes the erythromycin resistance gene, erythromycin or clarithromycin could be used as selection markers [46,47,48,49]. In the case of pJIR750ai, it has been used not only in *Clostridium* species but also in *Acetobacterium woodii* and *Eubacterium limosum*. The vectors contain both pIP404 and ColE1 of gram-positive and gram-negative bacteria origin, respectively. The antibiotic resistance gene is *catP*, which encodes for chloramphenicol acetyltransferase; hence, chloramphenicol and thiamphenicol can be used as selection markers [44,50,51,52,53]. In addition to these two plasmids, the pMTL80000 modular vector series was developed to transfer foreign DNA between *Clostridium* species and *E. coli*. These vectors contain plasmids of one of four gram-positive origins, pBP1, pCB10*2*, pIM13, and pCD6, and have p15a or ColE1 of gram-negative origins, and one of *ermB*, *catP*, or *tetA* as antibiotic resistance genes [54]. These vectors have been applied in *E. limosum, C. ljungdahlii*, and *C. autoethanogenum*; however, it was reported that their transformation efficiency in *E. limosum* is lower than that of the pJIR750ai vector [44].

### 2.2. Development of Genome Engineering Tools in Acetogens

The plasmid-based gene expression system has enabled the overexpression of native and heterologous genes for desired phenotypes. However, the instability of plasmids in hosts and the use of antibiotics for their maintenance can be problematic, especially in an industrial context [60,61,62]. To tackle these issues, integration of specific DNA fragments into the genome has been accomplished through HR, allowing for stable expression of the integrated genes. In addition, the development of several genome engineering tools such as ClosTron, transposon mutagenesis, and CRISPR-Cas systems, has also facilitated chromosomal deletion and insertion for metabolic engineering as well as physiological studies of acetogens (Table 2).

#### 2.2.1. ClosTron

ClosTron utilizes a mobile group II intron from the *ltrB* gene of *Lactococcus lactis*, which is integrated into the target genomic site with the aid of an intron-encoded protein possessing reverse transcriptase activity via an RNA-mediated retrohoming mechanism [65,66] (Figure 1A). Once the intron is inserted into a specific gene in the antisense orientation, the gene function can be disrupted. This tool has been widely demonstrated within the genus *Clostridium*, and hence it is termed “ClosTron” [67]. The well-established web-based ClosTron design tool streamlined the whole procedure [68,69]. Furthermore, selection markers simplified mutant isolation based on acquisition of antibiotic resistance (e.g., clarithromycin and thiamphenicol). Among acetogens, this tool has been implemented in *C. autoethanogenum* [28,36,37,38] and *C. ljungdahlii* [39]. Most of these studies utilized ClosTron to disrupt the genes involved in carbon fixation and energy conservation, which are the keys to understanding the basic physiology of acetogens. For instance, three isogenes encoding carbon monoxide dehydrogenase (CODH), *acsA, cooS1,* and *cooS2,* present in *C. autoethanogenum* were subjected to ClosTron-based gene inactivation. While *cooS1* or *cooS2* inactivation mutants exhibited no growth retardation effect under CO or H_2_/CO_2_, the *acsA* mutant could not grow on both autotrophic conditions. This result indicated that *acsA* is essential for carbon fixation in *C. autoethanogenum* [38]. Although the ClosTron system is a valuable tool for generating deletion mutants, its use is limited by the length of an integrated DNA sequence, as the efficiency decreases when the sequence length exceeds 1 kb [70] and its use leaves a scar of the inserted intron in the genome [68].

#### 2.2.2. Transposon Mutagenesis

Non-targeted gene integration can be accomplished with transposon mutagenesis, which randomly inserts DNA sequences into the host chromosome (Figure 1B). The conjugative transposons Tn*916* and Tn*925* from *Enterococcus faecalis* were the first elements exploited for random gene insertions in *A. woodii* [63]. The mariner-type *Himar1* from *Haematobia irritans* has been utilized as an alternative transposon to integrate large biosynthetic pathways that are difficult to insert with ClosTron, [71,72]. *Himar1* transposase inserts a DNA fragment flanked by inverted terminal repeats (ITRs) into a “TA” target site through a “cut-and-paste” mechanism. It has been reported to deliver biosynthetic pathways of up to 57.5 kb into the genome [73]. Recently, *Himar1* was successfully applied in *C. ljungdahlii* to integrate a 5 kb gene cluster (*adc, thlA, ctfA-ctfB* from *C. acetobutylicum*) for acetone production by conjugal transfer of donor plasmids [64].

#### 2.2.3. Homologous Recombination

Another strategy for stable genome insertion or deletion is to utilize HR. Double-crossover is typically used for this approach and occurs in two steps (Figure 1C). The first event is a single-crossover, in which the entire vector is incorporated into the target site through homology arms flanking a cargo sequence. Subsequently, the second crossover removes the region between the homologous sequences from the genome and leaves only the cargo DNA. This second recombination event occurs at very low frequencies in acetogens, which makes screening of the desired mutant cumbersome and time-consuming. To facilitate this, the method has been combined with the use of appropriate selectable markers. Antibiotic resistance genes, such as *ermB* and *catP,* are widely used in acetogens as positive selection markers that confer a selective advantage to the host, allowing only double-crossover mutants to survive under selective pressures. Based on this approach, earlier efforts on HR-based gene deletion were achieved in *C. ljungdahlii* [29,30]. Leang et al. [37], for instance, successfully deleted *adhE1* and *adhE2* bi-functional aldehyde/alcohol dehydrogenases for ethanol production, which led to less ethanol formation. Because of the limited number of selection markers available for *C. ljungdahlii* [30,31], the Cre-loxP system has been adopted to simultaneously disrupt three genes, *pta, adhE1*, and CLJU_c39430, by allowing reuse of antibiotic resistance genes [31].

On the other hand, counter-selectable markers, *pyrE* (orotate phophoribotransferase) and *pyrF* (orotate monophosphate decarboxylase), can be used as both positive and negative selection markers. The protocol to utilize these markers is typically performed in two steps. First, either *pyrE* or *pyrF* in the host is inactivated and creates uracil auxotrophs, which can be isolated by the addition of 5-fluoroorotic acid (5-FOA) as it is converted to a toxic compound in the presence of either enzyme. The mutant is subsequently complemented with a heterologous version of the respective gene included in a donor DNA plasmid. Consequently, double-crossover mutants can be readily isolated in uracil-free medium as they restore uracil protrophy [74]. This approach has been employed in *A. woodii* [25,26,27], *C. autoethanogenum* [28], *Moorella thermoacetica* [33], and *Thermoanaerobacter kivui* [34,35]. For example, a uracil auxotrophic mutant, ∆*pyrE*, was first generated in *A. woodii* to successfully delete *rnf* genes [25]. The authors used the heterologous *pyrE* gene from *C. acetobutylicum* to isolate the *rnf* deletion mutant that restored uracil prototrophy. In addition to generating a single deletion mutant, it is also possible to create double deletion mutants using the Allelic-Coupled Exchange (ACE), which couples a counter-selection marker gene to a desired double-crossover event [75]. Using the ACE strategy, Liew et al. [28] successfully constructed double mutants, *∆aor1+2* (isoforms of aldehyde:ferredoxin oxidoreductase) and *∆adhE1+2* (isoforms of bi-functional aldehyde/alcohol dehydrogenase) to prove that the indirect ethanol pathway comprising *aor* and *adh* is also critical in autotrophic ethanol production.

Besides chromosomal deletion to redirect metabolic fluxes, genomic integration of metabolic pathways consisting of several genes is desirable to acquire a stable industrial fermentation strain. HR-based insertion of large DNA fragments has been reported in *C. ljungdahlii* using either of two approaches: HR-based single-crossover on chromosomes [31] or heterologous phage attachment/integration (Att/Int) systems mediated by phase serine integrases [32]. These systems showed successful integration of the butyrate production pathway comprising eight genes (*thl, crt, bcd, etfB, etfA, hbd, ptb,* and *buk*).

#### 2.2.4. CRISPR-Cas Genome Editing

A multitude of genetic tools discussed above have accelerated metabolic engineering and the understanding of basic acetogen physiology. The majority of developed tools, however, still have some bottlenecks associated with a time-consuming mutant isolation process caused by low HR efficiency as well as inevitable scars left in the genome [30,31,36]. Recently, the CRISPR-Cas system has been extensively used as a tool for rapid, highly efficient, and markerless genetic editing in several model organisms [76,77,78]. Among the five CRISPR types [79], type II CRISPR-Cas9 from *Streptococcus pyogenes* is the best-characterized system and has been successfully applied in *C. autoethanogenum* [40], *C. ljungdahlii* [32,41,43], and *E. limosum* [44,45].

The CRISPR-Cas9 system consists of a Cas9 effector protein, which recognizes a 20 bp DNA sequence adjacent to the protospacer adjacent motif (PAM), “NGG”, guided by an RNA duplex composed of fused CRISPR RNA (crRNA) and trans-activating crRNA (tracrRNA) or engineered single guide RNA (sgRNA). When directed to a target site within the genome, Cas9 introduces double-stranded breaks (DSBs). In most prokaryotes, DSBs can be repaired by HR based on the donor DNA plasmid (Figure 1D). To fully exploit this system, Cas9 and sgRNA should be expressed at appropriate levels because Cas9 has intrinsic toxicity and the resulting DSBs can lead to cell death before recombination can occur. Due to the limited availability of well-established genetic parts or inducible promoters for acetogens, some studies have screened inducible promoters to tightly control Cas9 expression. For example, in *C. autoethanogenum*, the initial attempt to express Cas9 under a native strong constitutive promoter was not successful, likely due to toxicity caused by its uncontrolled expression. Thus, a library of tetracycline-inducible promoter variants was screened, and the selected synthetic inducible promoter, IPL12, fine-tuned Cas9 expression, leading to an improved editing efficiency of > 50% [40]. Similarly, several well-known inducible promoters were screened for Cas9 expression in *E. limosum*, and the anhydrotetracycline-inducible promoter was adopted for the CRISPR-Cas9 system in *E. limosum* [44]. Following its initial exemplification in several clostridial species [80], a synthetic, theophylline responsive riboswitch has also been exploited for Cas9-based mutant generation and complementation studies in *C. autoethanogenum* [81]. For the latter, unique 24-nucleotide “bookmark” sequences were incorporated into the mutant allele that acted as guide RNA targets during subsequent CRISPR-Cas9-mediated replacement with the complementing wildtype allele.

General bottlenecks for applying CRISPR-Cas systems in bacteria are the requirements of DNA cleavage, HR, and donor DNA. This renders its application even more difficult for acetogens, which are recalcitrant to foreign DNA uptake and have low recombination efficiencies [40,41,44,45]. Recently, an advanced genome editing tool at a one-nucleotide resolution, namely base editing, has been developed to circumvent these bottlenecks and was demonstrated in *C. ljungdahlii* [43]. This tool utilizes catalytically deactivated Cas9 (dCas9) fused with cytidine deaminase to generate cytosine-to-thymine substitutions. It requires only a gRNA cassette for targeting, without the need to induce lethal DSBs and offering donor DNA for recombination.

In addition to the most commonly used Cas9 protein, Cas12a (formerly known as Cpf1) [82] was also exploited in *C. ljungdahlii* to better meet its low GC content (ca. 43%). Cas12a recognizes AT-rich PAM sequences, “TTN”, which are more abundant than “NGG” for Cas9 in the genome [42]. One advantage of Cas12a is that its crRNA processing activity enhances the simplicity of multiplex genome editing [83,84]. To target multiple genes, Cas12a requires a single promoter and terminator to express multiple spacers and repeats, whereas Cas9 requires multiple promoters and terminators, which increase with the number of target genes [85]. Given the complexity of metabolic networks of the cell, it is imperative to modulate multiple genes rather than focusing on individual genetic manipulations. Therefore, using CRISPR for simultaneous deletion or insertion of multiple genes is worth considering when engineering acetogens for robust microbial cell factories that efficiently convert C1 to value-added biochemicals.

## 3. Engineering of Acetogens for Biochemical Production

As heterogeneous DNA transfer, plasmid-based genetic manipulation, and genome engineering tools have become established as useful techniques, studies on acetogens have largely been classified into three main strategies. The first is to increase C1 fixing efficiency, the second is to enhance the production of native biochemicals, and the last is to try to produce non-native biochemicals.

### 3.1. Improvement of C1-Fixing Pathways in Acetogens

As the plasmid-based gene expression system was developed, several studies have been conducted to express genes related to the WL pathway, energy production, or recycling pathways to improve the C1 gas utilization efficiency of acetogens.

#### 3.1.1. Engineering of the WL Pathway

The WL pathway consists of a methyl branch and a carbonyl branch (Figure 2) [3,5]. To enhance C1 fixation, genes encoding formyl-THF synthetase, methenyl-THF cyclohydrolase, methylene-THF dehydrogenase, and methylene-THF reductase of *C. ljungdahlii* were expressed in *A. woodii*. This mutant strain showed that acetate production was increased 1.2-fold, compared with the control strain under autotrophic growth conditions (Table 3) [50]. The carbon monoxide dehydrgogenase/acetyl-CoA synthase (CODH/ACS) complex plays an important role in the conversion of CO or CO_2_ to acetyl-CoA in acetogens [3,5]. When genes encoding CODH are disrupted, acetogens cannot grow under autotrophic conditions [38]. On the other hand, when the CODH/ACS gene (CAETHG_1620-1621) was overexpressed via the pMTL83157 plasmid in *C. autoethanogenum*, lag-phase growth decreased about 4.2 days and increased the production of ethanol by 1.2-fold or lactate by 2.7-fold (Table 3) [86]. These results indicate that the CODH/ACS complex is an engineering candidate to improve cell growth and chemical production in acetogens under autotrophic conditions.

#### 3.1.2. Discovery of Novel C1 Gas Fixing Pathways

In 2017, it was found that *Deltaproteobacteria* is capable of autotrophic growth under CO_2_ conditions via the glycine synthase-reductase pathway (GSRP) [87]. Among the acetogens, *Clostridium drakei* was found to have the GSRP, and it was demonstrated that the methyl branch of the WL pathway and GSRP can be used together to reduce CO_2_. According to these results, when the genes of *gcvTH, gcvPA/B, grdX, trxAB*, and *grdABCDE* constituting GSRP were introduced into *E. limosum*, which is known to lack GSRP, the growth rate and CO_2_ consumption rate of the GSRP-introduced strain increased 1.4-fold, and acetate production increased 2.1-fold compared to the wild-type strain (Figure 2 and Table 3) [52]. This result expanded the pioneering perspective for increasing the efficiency of C1 gas utilization by introducing a pathway to cooperate with the WL pathway in acetogens.

### 3.2. Production of Native Biochemicals

#### 3.2.1. Acetate

Acetogens can fix C1 gases to acetyl-CoA using one molecule of ATP via the WL pathway and then convert it to acetate through substrate-level phosphorylation by acetate kinase (ACK) to obtain one molecule of ATP (Figure 2). During the process of converting C1 gases into acetate, the net charge for ATP becomes zero. Therefore, acetate is the representative native product of most acetogens because the process of fixing C1 gases and producing acetate is a thermodynamically preferred chemical reaction (ΔG⁰′ = 9−0–180 kJ/mol) [88]. Improvement of acetate production has been achieved either by overexpressing acetate biosynthesis pathways or deleting other metabolite biosynthesis pathways that start from acetyl-CoA. In 2014, an engineered *A. woodii* strain was constructed to enhance acetate production by expressing both *pta* and *ack* genes encoding a phosphotransacetylase and acetate kinase from *C. ljungdahlii*, respectively, in a pJIR750ai backbone vector (Table 3) [50]. In *C. ljungdahlii*, when two genes, *adhE1* or *adhE2*, which encode putative bifunctional aldehyde/alcohol dehydrogenases, were deleted individually or together, acetate production was increased 1.7-fold (Table 2) [30].

#### 3.2.2. Ethanol

Ethanol can be produced from acetyl-CoA through two metabolic pathways in acetogens. The first pathway is catalyzed by an alcohol dehydrogenase (ADH)-dependent process using two NAD(P)Hs. The second pathway utilizes aldehyde-ferredoxin oxidoreductase (AOR) using reduced ferredoxin and NAD(P)H (Figure 2). In particular, *C. ljungdahlii*, *C. ragsdalei*, and *C. autoethanogenum* strains can produce ethanol as a native product, of which the Lanzatech process is a representative example of the successful commercial production of bio-ethanol from C1 gases [3,6,7,8,28,38,46,59,89]. Ethanol production was increased 1.5-fold by inducing the transcripts level of *adhE*1 genes in *C. ljungdahlii* [90]. Interestingly, ethanol production was specifically increased by 1.5~1.8-fold under CO autotrophic conditions when *adhE* genes were deleted in *C. autoethanogenum*. This indicates that using the AOR route is advantageous for ethanol production rather than the AdhE pathway in *C. autoethanogenum* under CO conditions (Table 2) [28,91]. Moreover, researchers expressed heat shock proteins such as GroEL, GroES, and DnaK in *C. ljungdahlii* or *C. autoethanogenum*, and its production was increased by enhancing resistance to the produced ethanol (Table 3) [49,92]. 

#### 3.2.3. 2,3-Butanediol (2,3-BDO)

*C. ljungdahlii*, *C. ragsdalei*, and *C. autoethanogenum* genomes encode genes for metabolic pathways that produce 2,3-BDO from pyruvate through acetoin. In particular, *C. autoethanogenum* could produce about 0.2 g/L of 2,3-BDO under batch culture conditions from CO steel mill gas [6,7,59]. To increase the production of 2,3-BDO, the *budC* gene encoding butanediol dehydrogenase of *Klebsiella pneumoniae* was expressed using the pMTL83155 vector in *C. autoethanogenum*. Subsequently, it produced about 0.37 g/L 2,3-BDO (Table 3) [93].

### 3.3. Production of Non-Native Biochemicals

#### 3.3.1. C3: Acetone or Isopropanol

ABE fermentation is a process for producing acetone, butanol, and ethanol, which is a representative industrial application of *Clostridium* strains including *C. acetobutylicum* [94,95]. Some efforts have been made to produce such non-native chemicals in other acetogens, such as *C. aceticum*, *C. ljungdahlii*, and *A. woodii* [47,51,64,90,96]. The acetone biosynthesis pathway begins when thiolase A (encoded by *thlA*) converts acetyl-CoA to acetoacetyl-CoA, which subsequently transfers CoA to acetate via CoA transferase (encoded by *ctfAB*) to produce acetoacetate. After removing CO_2_ by acetoacetate decarboxylase (encoded by *adc*), acetone is finally produced (Figure 2). This acetone biosynthesis pathway was constructed by expressing *thlA-ctfAB-adc* genes using pMTLs, pIMP, or pJIR vectors in several acetogens, which resulted in the production of 0.01–0.3 g/L acetone in *C. aceticum*, *C. ljungdahlii*, *C. autoethanogenum*, and *A. woodii* from C1 gases (Table 3) [47,51,96]. In addition, the acetone biosynthesis pathway was constructed using a lactose inducible promoter in *C. ljungdahlii*, and the transcriptional expression of the corresponding genes was induced to produce 0.1 g/L of acetone from C1 gases (Table 3) [90]. Isopropanol can be produced from acetone via alcohol dehydrogenases (encoded by *adh*) using the NAD(P)H cofactor. For example, 0.648 g/L of isopropanol was obtained from C1 gases when pMTL*-thlA-ctfAB-adc-adh2* was expressed in *C. autoethanogenum* (Table 3) [96].

#### 3.3.2. C4: *n*-Butanol or Butyrate

Butanol is another non-native product that cannot be produced by wild-type *C. ljungdahlii* and *C. autoethanogenum*. Therefore, studies have been performed to produce 1-butanol or 2-butanol by expressing the major butanol biosynthesis-related genes of *C. acetobutylicum* in the two acetogen strains. In the case of 1-butanol, thiolase (encoded by *thlA*) produces acetoacetyl-CoA from two acetyl-CoA, and 3-hydroxybutyryl-CoA dehydrogenase (encoded by *hbd*) is converted to 3-hydroxybutyryl-CoA. Subsequently, crotonase (encoded by *crt*) converts 3-hydroxybutyryl-CoA to crotonyl-CoA, which is then converted to butyryl-CoA by butyraldehyde dehydrogenase (encoded by *bcd*) and supported by an electron transfer flavoprotein (encoded by *etfAB*). Finally, 1-butanol is produced through butyraldehyde by butyraldehyde dehydrogenase using NAH(P)H (Figure 2 and Table 3). Alternatively, butyryl-CoA can be converted to butyrate via phosphotransbutylase (encoded by *ptb*) and butyrate kinase (encoded by *buk*), whilst also generating an ATP (Figure 2) [31,32,46,97]. This 1-butanol biosynthesis pathway was constructed in *C. ljungdahlii* by expressing the three genes *thlA, bdhA,* and *adhE* via plasmid, and 148.2 mg/L of 1-butanol and 70.5 mg/L of butyrate were produced from C1 gases (Table 3) [97]. The 2-butanol biosynthesis pathway consists of two reactions: meso-2,3-BDO is converted to 2-butanone (MEK) by propanediol/glycerol dehydratase (encoded by *pddABC*), and alcohol dehydrogenase (encoded by *adh*) is finally converted to 2-butanol using NAPDH (Figure 2). To produce 2-butanol in *C. autoethanogenum*, a 2-butanol biosynthesis pathway was constructed by expressing *als, aldc, budC, pddABC*, and *adh* genes, and the engineered strain produced 12.3 mg/L of 2-butanol (Table 3) [93].

#### 3.3.3. C5: Isoprene

Isoprene is one of the C5 terpenes and it can be produced from isopentenyl pyrophosphate (IPP). The IPP biosynthesis pathway is divided into two processes: the mevalonic acid (MVA) pathway synthesized from acetyl-CoA or the 1-deoxyxylulose 5-phosphate synthase (DXS) pathway synthesized from pyruvate and glycerol-3-phosphate (G3P) (Figure 2). Therefore, to produce isoprene from C1 gases, the MVA pathway or DXS pathway was introduced into *C. autoethanogenum* and *C. ljungdahlii*, respectively, and isopentenyl pyrophosphate isomerase (encoded by *idi*) and isoprene synthase (encoded by *ispS*) were additionally expressed [47,98]. These recombinant acetogens produced a total of 2 ng/mL of isoprene from syngas (Table 3) [56].

#### 3.3.4. Poly-3-hydroxybutyrate (PHB)

Poly-3-hydroxybutyrate (PHB) is a polyhydroxyalkanoate (PHA), a biodegradable plastic that is naturally produced from *Ralstonia eutropha* and other microorganisms, *including Alcaligenes*, *Pseuomonas*, *Bacillus*, *Rhodococcus*, *Staphylococcus*, and *Micrococcus* genus [99,100]. The PHB biosynthesis pathway converts acetyl-CoA to acetoacetyl-CoA, and then synthesizes 3-hydroxybutyryl-CoA using NAD(P)H. Subsequently, PHB can be produced through the polymerization of 3-hydroxybutyryl-CoA (Figure 2). To produce PHB from C1 gases using *C. autoethanogenum*, *phaA* (encoding a 3-ketothiolase), *phaB* (encoding an NAD(P)H-dependent acetoacetyl-CoA reductase), and *phaC* (encoding a PHA synthase) were expressed in *C. autoethanogenum*, and then this engineered strain produced approximately 22–27 mg/L of PHB from syngas (Table 3) [101].

So far, we have demonstrated the industrial potential of acetogens as biocatalysts through examples of various biochemicals produced from C1 gases. However, the low productivity of non-native biochemicals is not suitable for commercial scale production despite the production potentials [8]. To improve low productivity, it is necessary to construct genetic systems that can control the expression level of introduced foreign genes. Most of the genes involved in producing non-native biochemicals were regulated by native promoter/UTRs of *C. acetobutylicum, C. ljungdahlii, or C. autoethanogenum*. Since these promoter/UTR parts cannot quantitatively control the level of transcription or translation of target genes, the production of biochemicals using acetogens has limitations in the expansion from the lab scale to the commercial scale. Therefore, it is necessary to develop a regulatory system that can optimize the transcription or translation of each gene consisting of artificial pathways to efficiently synthesize biochemicals from C1 gases using acetogens.

## 4. Synthetic Biology Approach on Acetogens

In order to apply synthetic biology in the development of industrial platform microbes, synthetic parts (e.g., promoters, UTRs, and transcription terminators), which are the most basic synthetic biology units, must be developed first. To develop these parts, multi-omics information obtained from the genome, transcriptome, translatome, interactome, proteome, and metabolome of the acetogens of interest are required [17]. Based on these, we can not only understand their genetic regulation system under specific culture conditions but also design artificial genetic parts to regulate gene expression [14,15,104]. Synthetic biology is attracting attention from experts in many fields because the artificial organisms developed in this way could show many advanced behaviors that could not be controlled by simple gene expression such as gene deletion and overexpression. Therefore, synthetic biology provides a very powerful principle for gene expression and regulation by developing synthetic parts and the construction of genetic circuits.

### 4.1. Development of Synthetic Biology Tools in Acetogens

Many synthetic biology approaches for strain engineering are being attempted in acetogens, but research efforts are still limited compared to other industrial microbes. First, this is due to the limited understanding of the C1 fixation pathways of the acetogens and the genetic regulation system of many genes involved in their carbon and energy metabolisms. Second, there is a lack of efficient genetic manipulation tools. Third, high-throughput screening is not possible due to the lack of an efficient reporter system such as fluorescent proteins. Therefore, if the three limitations of acetogens are solved, more synthetic biology approaches will become applicable, and they can overcome the low productivity of using acetogens to convert C1 gases to biochemicals.

#### 4.1.1. Application of Multi-Omics Data with Genome-Scale Metabolic Models

With the rapid development of next-generation sequencing (NGS) technology, various omics data have been produced, including the transcriptome in *A. woodii* and *E. limosum* as well as *Clostridium* species such as *C. ljungdahlii, C. autoethanogenum,* and *C. drakei* (Table. 4). Through transcriptome analysis, it was possible to reveal the level of transcriptional expression of genes related to the WL pathway and the energy conservation system in acetogens under autotrophic conditions [36,105,106,107,108,109,110]. In addition, ribosome profiling in *C. ljungdahlii* or *E. limosum* revealed that translational regulation of genes in the carbonyl-branch or energy conservation system is critical to their autotrophic growth [111,112]. Recently, a genome-scale metabolic model (GEM) has been constructed for *C. ljungdahlii*, *C. autoethanogenum*, and *C. drakei* (Table 4). GEMs contain the entire set of known biochemical reactions taking place in an organism and allow us to predict, in silico, the flux distribution across the metabolic pathway. Based on in silico simulation/analysis using GEMs, the carbon flux or energy balance of acetogens during C1 fixation could be better understood [37,89,91,113,114,115,116,117]. This large amount of information based on systems biology can help understand the genetic regulation system for the C1 fixation process of acetogens.

#### 4.1.2. Development of Synthetic Promoters in Acetogens

Genetic promoter/UTR parts, which are a basic synthetic biology unit, allow the balance between growth and chemical production by quantitatively controlling the transcription and translation levels of target genes or pathways. However, the development of these genetic parts has been limited to acetogens. Using a robust native promoter such as thiolase A (P_thl_) or phosphotransacetylase/acetate kinase operon promoter (P_pta-ack_), the transcription level of the target gene in the plasmid was maintained constitutively high [39,51,59,98]. An inducible promoter system using the lactose-inducible promoter was applied to *C. ljungdahlii* and *C. autoethanogenum*, inducing the transcriptional expression level along with the lactose concentration [90].

In addition, information on the native promoter and 5’-UTR was revealed through differential RNA-seq in *E. limosum* and *A. bakii*. In particular, −35, −10, and ribosome-binding sites (RBS) of genes related to acetogenesis have strongly conserved “TTGACA”, “TATAAT’”, and “GGAGG” motifs [108,110]. This information could expand the understanding of the native promoter and 5′-UTR in acetogens, and succeed in developing promoter parts that can be regulated according to temperature [110]. However, these native parts may cause recombination problems with the genome because of their high sequence similarity with the genome. They are also affected by the intracellular transcriptional regulation system, so their functional orthogonality is limited. Therefore, synthetic promoter/UTR parts beyond the limitations of native parts are required. 

To develop synthetic parts, high-throughput screening methods using efficient reporter systems are required (Figure 3A). However, the reporter system based on fluorescence proteins that require oxygen has very low fluorescence intensity under anaerobic conditions [24,127,128], and FMN-based fluorescence proteins (LOV or FbFP) that can be used under anaerobic conditions are not sufficiently sensitive for overcoming the auto-fluorescence of several acetogens [129,130,131,132]. Due to these problems relating to reporter systems, the development of synthetic promoter/UTR genetic parts has not been conducted in acetogens. However, one study on the synthetic promoter/UTR was conducted in *C. ljungdahlii*. The promoter (P_thl_) of the acetyl-CoA acetyltransferase gene in the *C. acetobutylicum* genome was used as a backbone and a promoter/UTR library was constructed by randomization of sequences between −35 (TTG) and −10 (TATAAT) motifs, RBS (AGGAGG), and the start codon (ATG). The promoter library was connected upstream of the CatP-LacZ dual reporter, introduced into *C. acetobutylicum*, and screened according to the concentrations of chloramphenicol and LacZ to find synthetic promoter/UTR parts with 10-fold higher transcriptional activity than native P_thl_. After applying the genetic parts to *C. ljungdahlii*, it succeeded in increasing the productivity of isopropanol 19-fold compared to wild-type P_thl_ [49]. Indeed, this study demonstrated the capability of synthetic promoter/UTR parts in strain engineering to enhance biochemical productivity in acetogens. 

In model microbes, including *E. coli*, biosensors that can recognize specific chemicals or environmental stimuli have been developed, and many of these have been employed for high throughput screening of enzymes or mutant libraries according to their activities using flow cytometry [104,133,134,135]. However, the major bottleneck for the application of this approach to acetogens is the absence of efficient reporter systems. Thus, if efficient reporter systems are established, more complex genetic circuits can be designed and applied to acetogens. Moreover, these will enable the development of high throughput screening systems for obtaining better biochemical productivity or enzyme activity of acetogens.

### 4.2. Transcriptional Regulation System Based on CRISPR-Cas in Acetogens

#### 4.2.1. Repurposing CRISPR-Cas System for Metabolic Engineering

Repurposing the CRISPR-Cas system for transcription regulation is a genetic tool to knockdown or activate gene expression [18]. CRISPR interference (CRISPRi) replaces Cas9 with dCas9, which binds to a target site without cleaving the DNA, which consequently abolishes transcription [18,136]. Although CRISPR activation (CRISPRa) activates targeted gene expression with the assistance of transcription activators in other organisms [137,138], a lack of well-characterized activators suitable for bacteria has impeded CRISPRa applications in non-model organisms. In contrast, CRISPRi has shown great potential in metabolic pathway optimization to enhance the production of target products in several industrially important bacteria [139,140,141,142,143,144] (Figure 3B). Wu et al. [142] exploited CRISPRi in *E. coli* to suppress competing genes that could divert the carbon flux towards the production of 1,4-Butanediol. A similar approach has been shown in *C. ljungdahlii*, where they redirected acetyl-CoA flux from acetate to increase the production of 3-hydroxybutyric acid (3HB) [145]. Likewise, an ethanol synthesis gene (*adhE1*) was repressed with dCas12a instead of dCas9, and the redistributed carbon flux improved the titer of butyric acid [42].

#### 4.2.2. Genome-Wide CRISPRi/a Screening for Functional Genomic Studies

CRISPRi and CRISPRa techniques can be scaled up to the genome level. By introducing pooled gRNA libraries that can target nearly all genes, genome-wide CRISPR screening has enabled high-throughput functional genomics studies in bacteria [146,147,148,149] (Figure 3C). This approach can be used to identify essential genes associated with important physiological and metabolic processes in the cell. In acetogens, there are still large gaps in understanding the functions of unknown genes and putative essential genes, despite many earlier biochemical and mutational studies investigating important enzymes in the WL pathway and energy conservation pathways [4,25,29,36,38]. Using the genome-wide CRISPRi strategy, rapid genotype-to-phenotype mapping can be accomplished in acetogens. For example, Shin et al. conducted competitive growth assays with CRISPRi using sgRNA pools targeting five genes in *E. limosum* and showed that the number of cells harboring sgRNA, which target essential genes and hence decrease cell viability, was reduced from the initial population [44]. This study suggests the potential of genome-wide CRISPRi screening of essential and non-essential genes under autotrophic growth in acetogens.

#### 4.2.3. CRISPRi-Based Synthetic Genetic Circuits

Recently, CRISPRi-based synthetic genetic circuits have been designed and constructed to control cellular behaviors to fulfill the goals of synthetic biology (Figure 3D). Nielsen et al. [150] constructed multiple transcriptional logic gates utilizing CRISPRi, and the output of the logic circuits was used to control cellular phenotypes, such as sugar utilization or chemotaxis in *E. coli*. In addition, Cress et al. [151] constructed a library of orthogonally repressible promoters and it was incorporated into the branched violacein biosynthesis pathway as dCas9-dependent switches capable of redirecting carbon flux in *E. coli.* Employing these strategies in acetogens will allow the generation of better microbial cell factories that can control their regulatory and metabolic pathways in response to changing environments such as pH fluctuations and substrate availability.

## 5. Conclusions and Future Perspectives

The use of acetogens as biocatalysts in refining C1 gases needs to overcome several limitations in order to adapt them to industrial-scale applications. These limitations include low growth rates, toxicity by C1 gases and other substrates, and low productivity. The synthetic biology approach is expected to be a very powerful way to overcome these limitations. Along with the development of diverse bioparts and efficient genetic manipulation tools to build the targeted platform strains, the engineered acetogens should be further optimized under the conditions of interest. To this end, the adaptive evolution method is straightforward and robust. Mutations are naturally generated during adaptive evolution when a microbial strain is continuously exposed to a specific stress condition. If a genetic mutation effectively relieves a specific stress, the strain with the mutation shows fast growth under the specific stress conditions. There are some examples in which tolerance and growth rate of acetogen have been improved through multiple passages under growth conditions such as methanol or CO [53,152]. If adaptive evolution is additionally performed on acetogens, genetic variations related to optimize the growth rate or chemical production of acetogen under stress environments can be isolated efficiently.

Currently, there are not many examples of synthetic biology approaches for acetogens. However, due to the recent increase in data availability for systems biology approaches and efficient genetic manipulation tools such as CRISPR-Cas being developed, many obstacles to the synthetic biology approach are being resolved, and therefore these tools are now rapidly being adopted to discover gene modifications for strain optimization. Furthermore, if optimized acetogen genomes capable of highly efficient biochemical conversion of C1 gases are built, it is expected that the currently low productivity will be overcome, and industrial applications will increase.

## Figures and Tables

**Figure 1 ijms-21-07639-f001:**
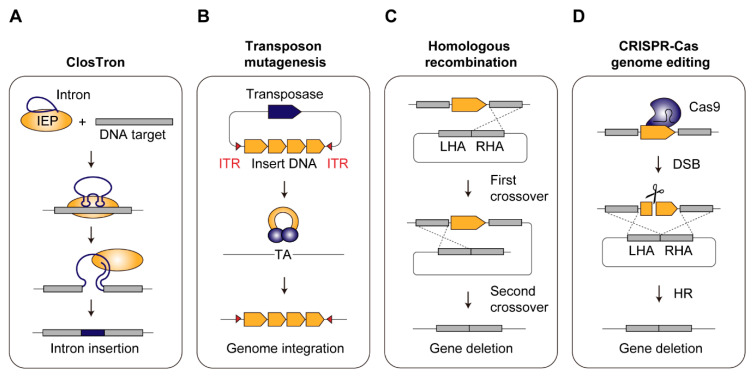
Genome engineering tools used in acetogens. (**A**). ClosTron is based on a complex form of an intron and intron-encoded protein and it inserts the intron into the target site in the genome. (**B**). Transposon mutagenesis inserts DNA sequences flanked by inverted terminal repeats (ITRs) into the genome by randomly recognizing a “TA” site in the genome. (**C**). Double-crossover homologous recombination occurs in two steps. The first crossover incorporates the entire vector into the genome and the subsequent second crossover removes the region between the left homology arm (LHA) and the right homology arm (RHA). (**D**). CRISPR-Cas system utilizes Cas9 and gRNA, which induces a double-stranded break (DSB) in the genome. The DSB can be repaired by homologous recombination between donor DNA and the host chromosome.

**Figure 2 ijms-21-07639-f002:**
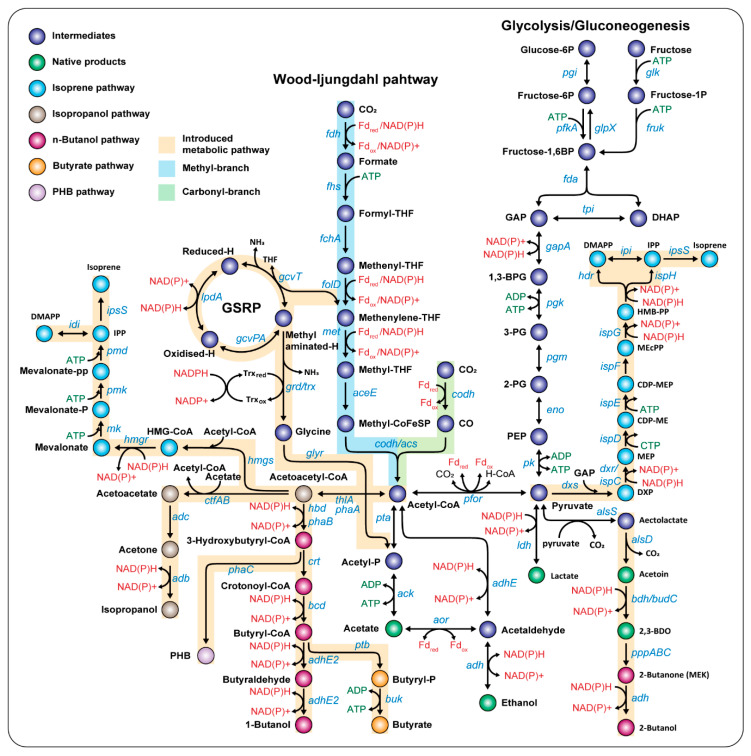
Metabolic pathways for engineering acetogens.

**Figure 3 ijms-21-07639-f003:**
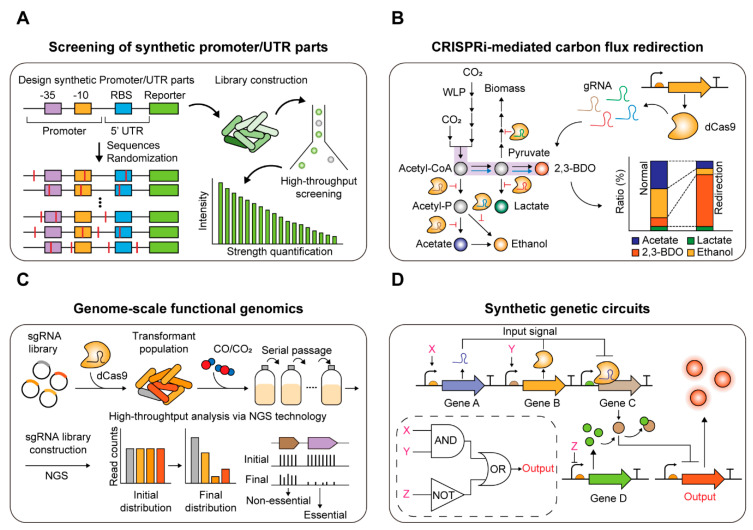
Synthetic biology approach to develop chassis microbe strain. (**A**) High-throughput screening of synthetic promoter/UTR parts. (**B**) Carbon flux redirection towards desired products through CRISPRi-mediated repression of genes in competing pathways. (**C**) Genome-wide CRISPRi/a screening for functional genomics studies. (**D**) Introduction of synthetic genetic circuits; Genetic regulation system using AND, NOT, and OR gate.

**Table 1 ijms-21-07639-t001:** Plasmid systems applicable to acetogens.

Plasmid	Gram (+) Replicon	Marker	Gram (−) Replicon	Applicable Species	Ref.
pIMP1	pIM13 (*repL*)	*amp/ermC*	ColE1	*Clostridium aceticum*, *Clostridium ljungdahlii*	[46,47]
pJIR750ai	pIP404	*catP*	ColE1	*Acetobacterium woodii*, *Eubacterium limosum*, *Clostridium ljungdahlii*	[44,50,52,53]
pK18mobsacB *	RP4	*catP/ermB*	pUC	*Moorella thermoacetica*	[33,55]
pMTL82151	pBP1 (*repA*)	*catP*	ColE1	*Clostridium ljungdahlii*, *Clostridium autoethanogenum*, *Eubacterium limosum*	[24,30,44,54]
pMTL82254	pBP1 (*repA*)	*ermB*	ColE1	*Clostridium ljungdahlii*, *Clostridium autoethanogenum*, *Eubacterium limosum*	[44,54]
pMTL83151	pCB102 (*repH*)	*catP*	ColE1	*Clostridium ljungdahlii*, *Clostridium autoethanogenum*, *Eubacterium limosum*	[24,28,44,54,56]
pMTL83245	pIM13 (*repL*)	*ermB*	ColE1	*Clostridium ljungdahlii*, *Clostridium autoethanogenum*	[54,57]
pMTL83353	pCB102 (*repH*)	*aad9*	ColE1	*Clostridium ljungdahlii*, *Clostridium autoethanogenum*	[54]
pMTL84151	pCD6 (*repA*)	*catP*	ColE1	*Clostridium ljungdahlii*, *Clostridium autoethanogenum*, *Eubacterium limosum*	[28,44,51,54]
pMTL84422	pCD6 (*repA*)	*tetA(P)*	p15a	*Clostridium ljungdahlii*, *Clostridium autoethanogenum*	[54]
pMTL85151	pIM13 (*repL*)	*catP*	ColE1	*Clostridium ljungdahlii*, *Clostridium autoethanogenum*, *Eubacterium limosum*	[44,54,58]
pMTL85241	pIM13 (*repL*)	*ermB*	ColE1	*Clostridium ljungdahlii*, *Clostridium autoethanogenum*	[54,59]

* modified vector; *ermC*, *ermB*, erythromycin (or clarithromycin) resistance gene; *catP*, chloramphenicol (thiamphenicol) resistance gene; *aad9*, spectinomycin resistance gene; *tetA(P)*, tetracycline resistance gene.

**Table 2 ijms-21-07639-t002:** Genome engineering tools used in acetogens.

Species	Genetic Manipulations	Ref.
**ClosTron**		
*C* *lostridium* *autoethanogenum*	Disruption of [FeFe]-hydrogenase and [NiFe]-hydrogenase genes involved in energy conservation	[36]
*C* *lostridium* *autoethanogenum*	Disruption of PCK, GAPDH, and Nfn complex genes	[37]
*C* *lostridium* *autoethanogenum*	Disruption of *acsA*, *cooS1*, and *cooS2* involved in carbon fixation	[38]
*C* *lostridium* *autoethanogenum*	Disruption of *adhE1*, *adhE2*, *aor1*, and *aor2* improved ethanol production to 180%	[28]
*C* *lostridium* *ljungdahlii*	Disruption of *adhE1* reduced ethanol production	[39]
**Transposon mutagenesis**		
*Acetobacterium woodii*	Insertion of Tn*925* or Tn*916* from *E. faecalis*	[63]
*C* *lostridium* *ljungdahlii*	Insertion of 5-kb acetone biosynthesis pathway via *Himar1* transposase	[64]
**Homologous recombination**		
*Acetobacterium woodii*	Deletion of ~5-kb *rnfCDGEAB* by HR in *∆pyrE* mutant generated by allelic-coupled exchange	[25]
*Acetobacterium woodii*	Deletion of ~5-kb *lctCDEF* by HR in *∆pyrE* mutant	[26]
*Acetobacterium woodii*	Deletion of ~3-kb *hydBA* by HR in *∆pyrE* mutant	[27]
*C* *lostridium* *autoethanogenum*	Double deletion of two *aor* or *adhE* isoforms via allelic exchange	[28]
*C* *lostridium* *ljungdahlii*	Deletion of *rnfAB* by HR	[29]
*C* *lostridium* *ljungdahlii*	Deletion of *fliA* via double-crossover; Deletion of *adhE1* and *adhE2* reduced ethanol production	[30]
*C* *lostridium* *ljungdahlii*	Insertion of a butyrate production pathway by HR; Deletion of *pta*, *adhE1* and CLJU_c39430 via Cre-loxP system redirected carbon and electron flux from acetate to butyrate synthesis	[31]
*C* *lostridium* *ljungdahlii*	Insertion of a butyric acid production pathway by phage serine integrase	[32]
*Moorella* *thermoacetica*	Insertion of *ldh* by HR in *∆pyrF* mutant	[33]
*Thermoanaerobacter kivui*	Deletion of *fruK* in *∆pyrE* mutant	[34]
*Thermoanaerobacter kivui*	Deletion of *fdhF*, *hycB3*, *hycB4*, and *hydA2*	[35]
**CRISPR-Cas**		
*C* *lostridium* *autoethanogenum*	SpCas9-mediated deletion of *adh* and *2*,*3-bdh*	[40]
*C* *lostridium* *ljungdahlii*	SpCas9-mediated deletion of *pta*, *adhE1*, *ctf* and *pyrE*	[41]
*C* *lostridium* *ljungdahlii*	SpCas9-mediated insertion of *attB* site and elimination of specific sites	[32]
*C* *lostridium* *ljungdahlii*	FnCas12a-mediated deletion of *pyrE*, *pta*, *adhE1*, and *ctf*	[42]
*C* *lostridium* *ljungdahlii*	Single nucleotide substitution of *pta*, *adhE1*, *adhE2*, *aor1*, and *aor2* using dCas9 fused with cytidine deaminase	[43]
*Eubacterium limosum*	SpCas9-mediated insertion of *ermB* gene into *folD* and *acsC* for gene disruption	[44]
*Eubacterium limosum*	SpCas9-mediated deletion of *pyrF*	[45]

**Table 3 ijms-21-07639-t003:** Biochemical production using the engineered acetogens.

Species	Plasmids	Genes	Product	Ref.
*C* *lostridium* *ljungdahlii*	pIMP1, pSOBPptb	*thlA*, *bdhA*, *adhE*	Butanol	[46]
*C* *lostridium* *ljungdahlii*	pMTL85246	*groES*, *groEL*	Ethanol	[92]
*C* *lostridium* *autoethanogenum, C* *lostridium* *ljungdahlii*	pMTL85245	*thlA*, *crt*, *bhd*, *bcd*, *etfAB*	Butanol	[97]
*Moorella thermoacetica*	pBAD, pK18	*ldh*, *dpyrF*, *g3pd*	Lactate	[33]
*C* *lostridium* *aceticum*	pIMP1	*thlA*, *ctfA/B*, *adc*, *atoDA*, *teII*, *ybgC*	Acetone	[47]
*C**lostridium**autoethanogenum*, *C**lostridium**ljungdahlii*	pMTL85147	*thla*, *ctfA/B*, *adc*	Acetone, Isopropanol	[96]
*C* *lostridium* *autoethanogenum*	pMTL85245, pMTL83245	malonyl-CoA reductase	3-HP	[57]
*C* *lostridium* *autoethanogenum*	pMTL85145, pMTL83155	*als*, *aldc*, *budC*, *adh*, *pddABC*	2,3-BDO, 2-Butanol	[93]
*C* *lostridium* *autoethanogenum*	pMTL85146, pMTL85246	*ispS*, *idi*, *dxs*, *hmgs*, *mk*, *pmk*, *pmd*	mevalonate, isoprene	[98]
*C* *lostridium* *autoethanogenum*	pMTL85141, pMTL85241	*budA*, *adh*, *alsS*	Acetoin, 2,3-BDO	[59]
*Acetobacterium* *woodii*	pJIR750ai	*pta*, *ack*, *thf*	Acetate	[50]
*C* *lostridium* *ljungdahlii*	pAH2, pKRAH1, pCL2, pJIR-*ermB*	*adhE1, adhE2, thlA-ctfAB-adc*	Ethanol, acetate, Acetone	[90]
*C* *lostridium* *ljungdahlii*	pMCSs, pJF100s, pDWs	*ispS*, *idi*	mevalonate, isoprene	[102]
*Acetobacterium* *woodii*	pMTL84151, pJIR750ai	*thlA*, *ctfA/B*, *adc*	Acetone	[51]
*C* *lostridium* *autoethanogenum*	pMTL83157	*codh/acs*	Ethanol, lactate	[86]
*C* *lostridium* *ljungdahlii*	pIMP1, pXY1	*thl*, *dnaK*	Ethanol, acetate	[49]
*Moorella thermoacetica*	pK18	*aldh*	Ethanol, acetate	[55]
*C* *lostridium* *ljungdahlii*	pMTL83151, pJF100s	*ispS*, *idi*	mevalonate, isoprene	[56]
*C* *lostridium* *autoethanogenum*	pMTL83157	*phaA*, *phaB*, *phaC*	PHB	[101]
*C**lostridium**ljungdahlii*, *C**lostridium**autoethanogenum*	pMTLs, pANNE99	*panBCD*, *bioBDF*, *bioHCA*	Pantothenate, Biotin, Thiamine	[103]
*Acetobacterium* *woodii*	pMTL83151	*gusA*, *arcABCD*	Ornithine	[103]
*Eubacterium limosum*	pJIR750ai	*gcvTH*, *gcvPA/B*, *grdX*, *trxAB*, *grdABCDE*	Acetate	[52]
*Eubacterium limosum*	pJIR750ai	*alsSD*	Acetoin	[53]

**Table 4 ijms-21-07639-t004:** System-level analysis of acetogens.

Year	Specices	Omics-Study	Ref.
2011.	*C**lostridium**ljungdahlii*, *C**lostridium**autoethanogenum*, *C**lostridium**ragsdalei*	Genome	[7]
2013	*C* *lostridium* *ljungdahlii*	GEMs	[118]
2013	*C* *lostridium* *ljungdahlii*	Transcriptome	[105]
2014	*C* *lostridium* *autoethanogenum*	Genome	[119]
2015	*Clostridium aceticum*	Genome	[120]
2015	*C* *lostridium* *autoethanogenum*	Transcriptome	[36]
2015	*C* *lostridium* *autoethanogenum*	Genome, GEMs, transcriptomics, metabolomics, Proteomics	[37]
2015	*C* *lostridium* *autoethanogenum*	Transcriptome	[106]
2015	*C* *lostridium* *autoethanogenum*	Genome	[121]
2015	*C* *lostridium* *ljungdahlii*	Transcriptome	[107]
2016	14 species	Pan-Genome	[122]
2016	*C* *lostridium* *ljungdahlii*	GEMs	[89]
2016	*C**lostridium**ljungdahlii*, *C**lostridium**autoethanogenum*, *Clostridium ragsdalei*, *Clostridium coskatii*	Comparison of genome	[39]
2017	*Eubacterium limosum*	Genome, TSS	[108]
2017	*Clostridium autoethanogenum*	GEMs, transcriptome	[113]
2017	*Clostridium ljungdahlii*	Transcriptome	[123]
2018	*A* *cetobacterium* *bakii*	Genome, TSS, Transcriptome	[110]
2018	*Eubacterium limosum*	Transcriptome, Translatome	[112]
2018	*A* *cetobacterium* *woodii*	Transcriptome	[109]
2018	*Clostridium ljungdahlii*	Translatome	[111]
2018	*Clostridium autoethanogenum*	GEMs, Proteomics, Metabolomics	[124]
2019	*A**cetobacterium**paludosum*, *A**cetobacterium**tundrae*, *A**cetobacterium**bakii*, *Alkalibaculum bacchi*	Comparison of genome	[125]
2019	*Clostridium autoethanogenum*	GEMs	[114]
2019	*Clostridium ljungdahlii*	GEMs, Proteomics	[115]
2019	*Clostridium* sp. AWRP	Genome	[126]
2020	*Clostridium drakei*	Genome, Transcriptome, GEMs	[52]
2020	*Clostridium autoethanogenum*	GEMs, Proteomics, Metabolomics	[91]
2020	*Clostridium ljungdahlii*	GEMs, Transcriptome	[116]
2020	*Clostridium autoethanogenum*	GEMs	[117]

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
