# Peer review of "Synthetic Biology on Acetogenic Bacteria for Highly Efficient Conversion of C1 Gases to Biochemicals"

_ijms, 2020, doi:10.3390/ijms21207639_

Round 1

Reviewer 1 Report

The authors have written a very comprehensive Review paper on acetogens while citing the state-of-the-art in this field while also presenting the bottlenecks.

There are a couple of minor points:

1) Lines 338-339 (and may be at other places): The names are actually genus and not species.

2) In section 4.1, it has been rightly pointed out what are the bottlenecks and absence of reporter is actually one of them resulting in absence of high-throughput screening methods.. The authors can also mention what is the state-of-the-art in Testing, and if there are examples of flow cytometry based screening with acetogens and if there are examples of biosensors developed in acetogens. Several examples now exist with other non-model organisms where biosensors have revolutionized the "testing" of strains for improved productivity or enzymes. What would be needed to have similar capabilities developed for acetogens.

Author Response

Point by point response

Reviewer #1:

Q1-1: Lines 338-339 (and may be at other places): The names are actually genus and not species.

A1-1: As the reviewer mentioned, the names of microorganisms were replaced with ‘genus’ in the modified manuscript.

Q1-2: In section 4.1, it has been rightly pointed out what are the bottlenecks and absence of reporter is actually one of them resulting in absence of high-throughput screening methods. The authors can also mention what is the state-of-the-art in Testing, and if there are examples of flow cytometry-based screening with acetogens and if there are examples of biosensors developed in acetogens. Several examples now exist with other non-model organisms where biosensors have revolutionized the "testing" of strains for improved productivity or enzymes. What would be needed to have similar capabilities developed for acetogens.

A1-2: As the reviewer pointed out, there are no biosensor studies for high-throughput testing on acetogens. The major bottleneck to this is the absence of efficient reporter systems. Thus, if efficient reporter systems are developed, the biosensor circuits can be developed for acetogens and high throughput screening will be possible. To discuss about this, the authors added sentences as following:

On page 15, line 444 to 449: “In model microbes, including E. coli, biosensors that can recognize specific chemicals or environmental stimuli have been developed, and many of these have been employed for high throughput screening of enzymes or mutant libraries according to their activities using flow cytometry [104,133-135]. However, the major bottleneck for the application of this approach to acetogens is the absence of efficient reporter systems. Thus, if efficient reporter systems are established, more complex genetic circuits can be designed and applied to acetogens. Also, these will enable to the development of high throughput screening systems for obtaining better biochemical productivity or enzyme activity of acetogens.”

Reviewer 2 Report

The manuscript entitled “Syntethic biology on acetogenic bacteria for highly efficient conversion of C1 gases to biochemcials” is well written and well organized. However, in my opinion, this paper is a bit general and, in some of the topics, should be better explored and detailed, namely, 3.2 (3.2.1, 3.2.2, 3.2.3).

Other corrections/suggestions that should be taking in account by the authors:

  • In the abstract the authors referred that synthesis gas is mainly produced from fossil fuels or biomass gasification which is correct; but, in the introduction and throughout the text, the authors refer to syngas only as a recycle waste gases from industries. Should be coherent.
  • Line 90, is the first time that abbreviation HR appears – must be written in full.
  • Lines 179 and 180 – the authors wrote here the mean of CRISPR, but, some lines before (line 91) you refer already the “CRISPR”. Please include the definition on line 91 or in both places.
  • The legend of the figures must follow the same format. In Figure 1 the letters used are uppercase and the different parts of the figure are identified as: A. XXXXX. In Figure 3, the authors should use the same formatting, instead of using lowercase letters and with brackets. Additionally, please correct the name of Figure 3, which in the text is referred to as Figure 3, but in the figure is mistakenly identified as Figure 1.
  • Line 405 – to be consistent with all the text, “Acetobacterium bakii” should be “A. bakii”.
  • In all tables, the names of the microorganisms, at least the first time they appear, the respective genus must be written in full. For Clostridium, there is no confusion, but for A. bakii and A. bacchi, which are different genera, the authors are misleading the reader. The tables get bigger and visually they may not look so beautiful, but they are scientifically clearer, more correct and better fulfill their objective information function. Please correct.
  • Still regarding Table 1, there is no need to include abbreviations for microorganisms (column: Applicable Strains), especially when they are used only in that table. Standardizing the way in which information is presented is the best choice. Please correct.

Author Response

Reviewer: #2:

Comment: The manuscript entitled “Synthetic biology on acetogenic bacteria for highly efficient conversion of C1 gases to biochemicals” is well written and well organized. However, in my opinion, this paper is a bit general and, in some of the topics, should be better explored and detailed, namely, 3.2 (3.2.1, 3.2.2, 3.2.3).

Response: According to the comment, the authors added some examples to the 3.2 section and reinforced it in more detail.

Q2-1: In the abstract the authors referred that synthesis gas is mainly produced from fossil fuels or biomass gasification which is correct; but, in the introduction and throughout the text, the authors refer to syngas only as a recycle waste gases from industries. Should be coherent.

A2-1: As the reviewer pointed out, the authors have modified the sentence as following:

On page 1, line 40-41: “Diverse acetogens have been suggested as promising biocatalysts to utilize C1 gases in synthesis gases or waste gases generated from industries using the WL pathway”

Q2-2: Line 90, is the first time that abbreviation HR appears – must be written in full.

A2-2: The authors first mentioned homologous recombination with its abbreviation HR in line 66.

Q2-3: Lines 179 and 180 – the authors wrote here the mean of CRISPR, but, some lines before (line 91) you refer already the “CRISPR”. Please include the definition on line 91 or in both places.

A2-3: As the reviewer pointed out, the authors have added the full name of CRISPR-Cas system in lines 66 and 67 where the word is first mentioned. The modified sentence is as following:

On page 3, line 66-67: “In addition, many metabolic engineering efforts have recently been made using homologous recombination (HR) [25-35], ClosTron [28,36-39] and Clustered Regularly Interspaced Short Palindromic Repeats (CRISPR) along with its CRISPR-associated (Cas) protein (CRISPR-Cas) system [32,40-45] to improve the production of value-added biochemicals from C1 gases.”

Q2-4: The legend of the figures must follow the same format. In Figure 1 the letters used are uppercase and the different parts of the figure are identified as: A. XXXXX. In Figure 3, the authors should use the same formatting, instead of using lowercase letters and with brackets. Additionally, please correct the name of Figure 3, which in the text is referred to as Figure 3, but in the figure is mistakenly identified as Figure 1.

A2-4: As the reviewer pointed out, the legend of Figure 3 was investigated and updated in the modified manuscript.

Q2-5: Line 405 – to be consistent with all the text, “Acetobacterium bakii” should be “A. bakii”.

A2-6: Line 405 was modified as following:

On page 14, line 405: “In addition, information on the native promoter and 5'-UTR was revealed through differential RNA-seq in E. limosum and A. bakii.”

Q2-6: In all tables, the names of the microorganisms, at least the first time they appear, the respective genus must be written in full. For Clostridium, there is no confusion, but for A. bakii and A. bacchi, which are different genera, the authors are misleading the reader. The tables get bigger and visually they may not look so beautiful, but they are scientifically clearer, more correct and better fulfill their objective information function. Please correct. Still regarding Table 1, there is no need to include abbreviations for microorganisms (column: Applicable Strains), especially when they are used only in that table. Standardizing the way in which information is presented is the best choice. Please correct.

A2-6: As the reviewer mentioned, the names of microorganisms in all tables were replaced with their full names in the modified manuscript.